# Improvement of CoCr Alloy Characteristics by Ti-Based Carbonitride Coatings Used in Orthopedic Applications

**Mihaela Dinu** [1], **Iulian Pana** [1], **Petronela Scripca** [1], **Ioan Gabriel Sandu** [2], **Catalin Vitelaru** [1] **and Alina Vladescu** [1,3,*]

1   Department for Advanced Surface Processing and Analysis by Vacuum Technologies, National Institute of Research and Development for Optoelectronics-INOE 2000, 409 Atomistilor St., 077125 Magurele, Romania; mihaela.dinu@inoe.ro (M.D.); iulian.pana@inoe.ro (I.P.); petronela.scripca@inoe.ro (P.S.); catalin.vitelaru@inoe.ro (C.V.)

2   Faculty of Material Sciences and Engineering, Gheorghe Asachi Technical University of Iasi, 41 Prof.dr.doc. D. Mangeron St., 700050 Iasi, Romania; gisandu@yahoo.com

3   Physical Materials Science and Composite Materials Centre, National Research Tomsk Polytechnic University, Lenin Avenue 43, 634050 Tomsk, Russia

*   Correspondence: alinava@inoe.ro; Tel.: +4-21-457-57-59

**Abstract:** The response of the human body to implanted biomaterials involves several complex reactions. The potential success of implantation depends on the knowledge of the interaction between the biomaterials and the corrosive environment prior to the implantation. Thus, in the present study, the in vitro corrosion behavior of biocompatible carbonitride-based coatings are discussed, based on microstructure, mechanical properties, roughness and morphology. TiCN and TiSiCN coatings were prepared by the cathodic arc deposition method and were analyzed as a possible solution for load bearing implants. It was found that both coatings have an almost stoichiometric structure, being solid solutions, which consist of a mixture of TiC and TiN, with a face-centered cubic (FCC) structure. The crystallite size decreased with the addition of Si into the TiCN matrix: the crystallite size of TiCN was 16.4 nm, while TiSiCN was 14.6 nm. The addition of Si into TiCN resulted in smaller $R_a$ roughness values, indicating a beneficial effect of Si. All investigated surfaces have positive skewness, being adequate for the load bearing implants, which work in a corrosive environment. The hardness of the TiCN coating was 36.6 ± 2.9 GPa and was significantly increased to 47.4 ± 1 GPa when small amounts of Si were added into the TiCN layer structure. A sharp increase in resistance to plastic deformation ($H^3/E^2$ ratio) from 0.63 to 1.1 was found after the addition of Si into the TiCN matrix. The most electropositive value of corrosion potential was found for the TiSiCN coating (−14 mV), as well as the smallest value of corrosion current density (49.6 nA cm$^2$), indicating good corrosion resistance in 90% DMEM + 10% FBS, at 37 ± 0.5 °C.

**Keywords:** titanium-based carbonitrides; coating; corrosion resistance; X-ray diffraction; nanoindentation; cathodic arc deposition

## 1. Introduction

In medical applications, especially for hip and knee implants, a CoCr alloy is mostly used, due to good mechanical, anticorrosive and tribological characteristics [1–3]. CoCr alloys have also been used as screws in trauma plating systems. In this case, a low osseointegration, compared to Ti alloys, could facilitate an easier removal after the healing of the bone fracture. Due to its higher strength, a CoCr alloy was also used for idiopathic scoliosis applications, where the results proved to be better in the

case of the CoCr alloy compared to stainless steel (SS) and a Ti alloy [4]. Another application was the use of a CoCr alloy in implants used for the correction of spine deformities, due to the high rigidity of the CoCr rod compared to SS- and Ti-based ones [5,6]. CoCr was also found to be well adapted in dentistry for its good castability, especially the wrought alloys. Guide wires, clips, orthodontic arch wires and catheters are among the main applications in this field [7,8]. Thus, the decreased corrosion resistance of SS and the low wear resistance of Ti alloys make it difficult to replace CoCr alloys in a wide range of applications.

Despite these good properties, in clinical practice the implants made of a CoCr alloy exhibited a high rate of failure due to various complications after implantation: (i) high toxicity as a result of the migration of toxic metal ions; (ii) a high amount of wear debris surrounding the peri-implant tissues and body organs, mainly due to the low wear resistance in biological fluids; (iii) low bioactivity abilities; (iv) a non-hydrophilic surface [1,2]. During the friction process, wear debris of CoCr alloys were generated in different size and shapes [9] and migrated to the periprosthetic tissues, leading to a failure of the implants. The wear of hip or knee joints is a complex process that involves many factors, such as the material and geometry of the implant, synovial fluid properties (various protein levels) and the patients' lifestyles and body weight. Thus, the wear particles larger than 0.5–10 μm (round to oval to irregular shapes) play a dramatic role as third-body wear, leading to an intense wear process [9–11]. The main problem with these debris is their high size, as cells (i.e., macrophages, fibroblasts, giant cells, neutrophils, lymphocytes, osteoclasts) will interact with these debris, leading to a chronic inflammatory response [9,12–14]. Another problem can be the release of Co and Cr ions into the synovial fluid and their correlated increased concentration in the blood. Although considered essential elements for the body, their increase in concentration can be detrimental for certain functions [15]. Thus, their long-term exposure can lead to cellular effects in the adjacent tissue and even to necrosis [16]. Nevertheless, in order to eliminate the mentioned disadvantages found in CoCr alloy medical implants, several modifications of alloy surfaces were carried out over recent decades by: (i) coating, using various types of techniques (PVD, CVD, ion implantation, plasma spray); (ii) surface structuring (laser processing, sand blasting, acid etching, anodization); (iii) micro arc oxidation; (iv) electrochemical oxidation [1,17]. The PVD method chosen for this study, namely the cathodic arc method, combines both a high degree of ionization of the ejected particles and a high efficiency of the evaporation process. Even though the initial energies are about 20 eV for light elements and around 200 eV for heavy elements, the final ion velocities (in the range of $1$–$2 \times 10^4$ m/s) were found to be independent of the cathode material and ion charge state, due to electron–ion coupling [18,19]. Thus, with an enhanced atom mobility and surface diffusion, due to the higher energy of the ions, the obtained materials have favorable conditions in order to obtain different coating properties [20].

The present study aimed to analyze TiSiCN as a possible coating solution to improve the corrosion and wear resistance of CoCr alloys used for orthopedic implants. For comparison, TiCN and uncoated CoCr were used as control groups. The TiCN coating was selected as a reference because it has good mechanical properties and good corrosion resistance, and an acceptable wear resistance in dry environments [21–23]. By the addition of Si into TiCN, it was expected to significantly reduce the friction and wear process, as well as to improve the corrosion resistance of the CoCr alloy. It was reported that the addition of Si governs grain refinement and Si-containing coatings present superior friction and wear performance [24]. Furthermore, it was demonstrated that the addition of Si to various materials with biological applications enhances the proliferation and differentiation of human osteoblasts, accelerating the osseointegration process [25]. Additionally, Si–N thin films proved to have remarkable properties, which included high thermal stability and chemical inertness, in addition to those already mentioned [26]. A survey of the literature shows that TiSiCN coatings have a superior tribological performance, but tests were performed mainly in conditions used in industrial applications, such as cutting tools and the automotive industry. Their main advantages are low friction, high wear resistance, good mechanical properties such as toughness and high resistance on plastic deformation [27–32]. For medical applications, however, they have not yet been tested. Nevertheless,

an alternative solution was the addition of Zr and Cr to the Si–N and Si–C–N matrix for severe wear and corrosive applications [33].

In the present study, the coatings were obtained by the cathodic arc evaporation method on the CoCr substrates under a mixture of $CH_4$ and $N_2$ gases. The investigation included the examination of elemental and phase composition, texture, structure, morphology and mechanical properties (Young modulus, hardness, roughness, stress). Special attention was devoted to the corrosion resistance performed in a 90% DMEM + 10% FBS solution, at $37 \pm 0.5\,°C$. In order to understand the damaging effect of the corrosion test, the morphology and roughness after corrosion were evaluated. Electrochemical impedance spectroscopy (EIS) was also performed in order to investigate the behavior of the proposed systems. Thus, this method gave an insight into the electrochemical processes which occurred at the material–electrolyte interfaces. The research was conducted in order to find new and improved structures as a better solution to optimize the safety and efficacy of biomaterials.

## 2. Materials and Methods

### 2.1. Coatings Preparation

Both coatings were prepared by the cathodic arc deposition process. For TiCN, one Ti cathode (99.99% purity) was used, while for TiSiCN, a Ti+Si cathode (85 at.% Ti and 15 at.% Si; 99.9% purity) was used. The cathodes were supplied from Cathay Advanced Materials Limited, Guangdong, China. The coatings were prepared using both $CH_4$ and $N_2$ as reactive gases (99.999% purity, Linde). The position of each cathode inside the deposition chamber was the same as those described in [33]. In order to guarantee the uniform thickness of the coatings, the substrate holder was rotated by 15 rot/min. The CoCr substrates (ASTM F75 CoCr alloy) were cut into 12 mm discs and polished up to Ra roughness of $46.9 \pm 5.9$ nm. Each substrate was ultrasonically cleaned in trichloroethylene and flushed with dry nitrogen, and then introduced in the deposition chamber. To eliminate any impurity, the substrates were sputter etched with $Ar^+$ ions (1 keV) for 15 min. The deposition parameters are listed in Table 1, and were maintained constant during all deposition runs. The same negative substrate bias voltage was applied on both cathodes and the substrate temperature was around 320 °C. The thickness of the coatings was around 2.5 μm.

**Table 1.** Conditions for the developed coatings.

| Deposition Parameters | TiCN | TiSiCN |
|---|:---:|:---:|
| Base pressure | $2 \times 10^{-3}$ Pa | |
| Working pressure | $2 \times 10^{-2}$ Pa | $6 \times 10^{-2}$ Pa |
| $CH_4$ mass flow rate | 80 sccm | |
| $N_2$ mass flow | 120 sccm | |
| Arc current on each cathode | 90 A | |
| Substrate bias voltage | −150 V | |
| Deposition duration | 40 min | |

### 2.2. Coatings Characterization

Energy dispersive spectrometry (EDS) was used for the analysis of the elemental composition (EDS, Quantax70, Bruker, Tokyo, Japan). The morphology was examined by scanning electron microscopy (SEM, Hitachi TM3030Plus, Tokyo, Japan). Phase composition was studied by the X-ray diffraction method (XRD, SmartLab diffractometer, Rigaku, Tokyo, Japan), using Cu Kα radiation, from 10° to 100° with a step size of 0.02°/min. For the thickness and surface roughness determination, a surface profilometer (Dektak 150, Bruker, Billerica, MA, USA) was used. Surface roughness was measured for each investigated sample on five line-scans, each on a distance of 4000 μm.

The mechanical properties of the coatings were determined using a Hysitron Premier TI nanoindentation unit equipped with a Berkovich indenter tip of 100 nm radius and a total included

angle of 142.3°, respectively. Prior to any sample testing, the Z-axis calibration was performed in the air and the machine compliance was assured using a fused quartz standard calibration sample with known hardness (H = 9.25 GPa ± 10%) and elastic modulus (E = 69.6 GPa ± 10%). In order to perform the nanoindentation experiments, a 15 × 15 μm² area was previously scanned using the same Berkovich diamond tip at a normal force of 2 μN to investigate the surface roughness for the subsequent indents. Additionally, the indents were intentionally located at least 5 μm apart from each other, whilst an applied force of 10 mN was employed for every nanoindentation test. The force–displacement curves were recorded using a gradual force increase up to 10 mN in a 7-s time interval, followed by a 2-s dwell time at the maximum force of 10 mN and a gradual force decrease within the next 7 s, until the complete tip retraction from the coatings surface.

The electrochemical behavior of the investigated specimens was analyzed by potentiodynamic polarization and electrochemical impedance spectroscopy (EIS), using a VersaSTAT 3 Potentiostat/ Galvanostat system. The measurements were performed in 90% DMEM + 10% FBS, at 37 ± 0.5 °C, using a typical three electrode setup with a Pt grid counter electrode (CE) and a saturated calomel (saturated KCl) (SCE) as the reference electrode (RE), while the working electrode consisted of uncoated CoCr and TiCN or TiSiCN coated CoCr substrates, respectively.

The open circuit potential ($E_{OC}$) was monitored for 1 h, starting after the sample's immersion. Linear polarization, Tafel and potentiodynamic curves were performed by applying a potential of −20 to 20 mV vs. $E_{OC}$, −50 to 250 mV vs. $E_{OC}$ and −1 V vs. $E_{OC}$ to 2 V vs. RE, respectively, with a scanning rate of 0.167 mV/s. For the linear polarization measurements, the testing conditions were selected based on preliminary results, in such a way that the applied potential gave a linear behavior. The selected value was used to accommodate all the investigated systems. The EIS measurements were performed over a range of frequencies ($0.1 \div 10^4$ Hz), by applying a sinusoidal signal of 10 mV RMS vs. $E_{OC}$.

## 3. Results

### 3.1. Elemental and Phase Composition

The elemental composition of the coatings obtained by the EDS method is presented in Table 2. Both coatings had an almost stoichiometric structure. The amount of Si was low but proved to be sufficient for the goal of this study. Both coatings exhibited a preferred growth orientation after (111) plan (Figure 1). The TiCN peaks were located between those of TiC and TiN, indicating that the TiCN is a solid solution and consisted of a mixture of TiC and TiN, with a face-centered cubic (FCC) structure (like NaCl). The same results were also reported by other authors regarding the TiCN coatings [21]. The identification of TiCN was performed according to JCPDS no. 042-1488 (red lines in Figure 1). It can be seen that our TiCN is shifted towards a lower Bragg angle compared with the TiCN standard, which is related to the formation of more TiC phase (Figure 1). This finding is in accordance with the results reported by Wang et al. [34] regarding the TiSiCN coatings. The preferred orientation of both coatings was also confirmed by the texture coefficient T(*hkl*), a strong texture intensity at (111) plane being found in the case of both coatings. Karlsson et al. reported that the plane with a preferred orientation parallel to the investigated surface has a texture coefficient value higher than 1 [21].

**Table 2.** Elemental composition measured by EDS, texture coefficient T(*hkl*), crystallite size *d* and strain *ε* determined by the Williamson–Hall plot method.

| Coating | Elemental Composition (at.%) | | | | (C+N)/ Σ(Me+Si) | T(*hkl*) | | | | d (nm) | ε |
|---------|------|-----|------|------|------|-------|-------|-------|-------|------|------|
| | Ti | Si | C | N | | (111) | (200) | (220) | (311) | | |
| TiCN | 52.1 | - | 17.5 | 30.4 | 0.9 | 0.59 | 0.09 | 0.24 | 0.07 | 16.4 | 0.053 |
| TiSiCN | 48.4 | 3.4 | 20.1 | 28.1 | 0.9 | 0.73 | 0.06 | 0.17 | 0.05 | 14.6 | 0.012 |

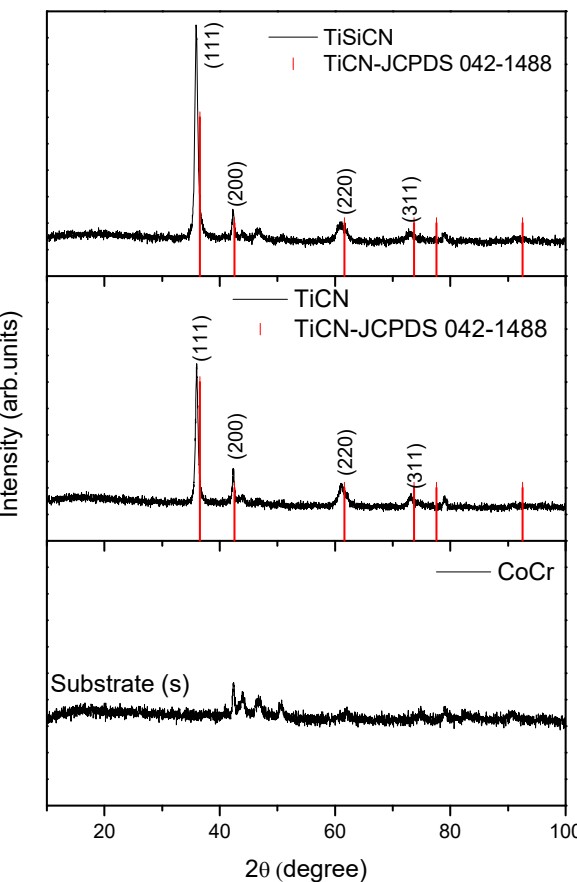

**Figure 1.** X-ray diffraction patterns of the investigated coatings (S-substrate) as well as of the TiCN standard (JCPDS 042-1488).

The crystallite size was determined by the Debye–Scherrer equation of the peak (111) and is presented in Table 2. The crystallite size decreased from 16.4 nm for TiCN coatings to 14.6 nm for TiSiCN. The possible explanation is the penetration of impinging Si ions into the lattice of the TiCN coating, and the decreased generation of defects, led to a decreased number of preferential nucleation sites causing reduced grain [35,36]. The strain $\varepsilon$ was determined by the Williamson–Hall plot method and is presented in Table 2. In both cases, the strain was low, but after the addition of Si, the strain was significantly reduced. It is reasonable to believe that the observed difference in strain of the TiSiCN coatings is related to the amorphous phases in which Si can be found (Si, SiNx, SiCN) or the C=C phase at grain boundaries, which is detrimental for crystallite development. These effects will be goal of another paper, in which other complex analyses, such as TEM and XPS, will be carried out.

The lattice parameter obtained from 2θ values of the (111), (200), (220) and (311) of TiSiCN was 4.2771, while for TiCN it was 4.2725. According to JCPDS no. 042-1488, TiCN has a lattice parameter equal to 4.2644. When the diffraction peak is shifted towards lower Bragg angle, an increase of lattice parameter is observed. This result is in good agreement with those of Constantin et al. related to TiCN coatings [37]. Both coatings exceeded the size of the standard unit cell parameters. This finding can be a sign that the hardness can be low and, on the other hand, the toughness will be high [38].

Grieveson et al. stated that the TiN and TiC compounds do not combine perfectly; the mixture is far from the ideal Raoultian behavior [39,40]. The same conclusion was also considered by Levi et al., who revealed that the occupation of all sites by Ti, C and N is not random, it is a TiN-based structure with a replacement of the N site by C, forming an FCC or tetragonal structures on the low concentration of vacancies that might be present [41]. All these aspects had an important effect on the characteristics of the TiCN coatings.

### 3.2. Morphology and Roughness (Before Corrosion)

The surface morphologies of the uncoated CoCr substrate and the TiCN and TiSiCN coatings can be observed in Figure 2. The appearance of the coatings is characterized by a continuous coverage of the substrate with visible individual microparticles distributed on the surface. These droplets are considered defects and are characteristic of the energetic ejection of the particles during the arc deposition process [42,43].

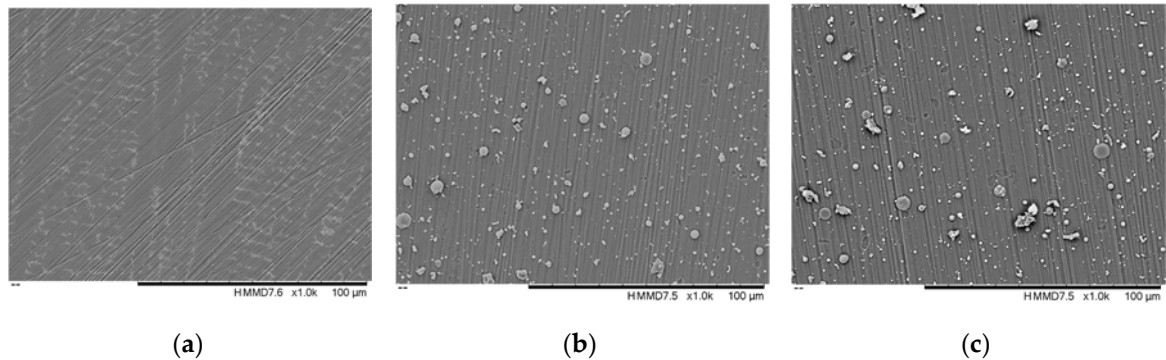

| (**a**) | (**b**) | (**c**) |

**Figure 2.** SEM images of the (**a**) uncoated CoCr substrate and (**b**) TiCN and (**c**) TiSiCN coatings.

### 3.3. Roughness of Surface

The main surface texture parameters of the investigated surfaces are: $R_a$: the arithmetic average of the roughness profile, $R_q$: the root mean square average of the roughness profile, $S_k$: skewness (according to the ISO 2517 standard [44]), and are presented in Table 3. The uncoated substates had an $R_a$ roughness around 50 μm, as intended. The roughness of the coated surfaces was much higher and it was more evident in the TiCN coated surfaces. Similar values of both $R_a$ and $R_q$ were found for the TiCN and TiSiCN coatings.

**Table 3.** Main roughness parameters of the investigated specimens.

| Sample | CoCr | TiCN | TiSiCN |
|---|---|---|---|
| $R_a$ (nm) | 46.9 ± 5.9 | 534.6 ± 111.4 | 499.7 ± 28.9 |
| $R_q$ (nm) | 59.5 ± 7.0 | 746.8 ± 181.9 | 650.4 ± 30.7 |
| $S_k$ | 0.2 ± 0.4 | 2.1 ± 0.3 | 1.6 ± 0.1 |

A negative value of $S_k$ shows that the surface consists of many valleys, while a surface with a positive $S_k$ contains mainly peaks and asperities. Therefore, a surface with positive $S_k$ has a good tribological performance in lubrication conditions. Taking into account this observation, both investigated coatings exhibited a positive $S_k$, which is suitable for load bearing implants, which work in a corrosive environment. The uncoated substrate had an $S_k$ close to 0, meaning that the surface was very flat. The TiCN surfaces had a high $S_k$, indicating many peaks and asperities, whereas Si-containing coatings had a smaller value, indicating a decrease in peaks and asperities. The dependence of corrosion resistance on these parameters will be discussed below.

### 3.4. Hardness and Elastic Modulus

Nanoindentation is a widely used technique, and it is a powerful tool to investigate the nanomechanical properties of materials, such as hardness and elasticity [45]. In order to overcome the substrate and roughness effects on nanoindentation tests, many studies have pointed out the importance of two basic rules. They refer to the maximum penetration depth, which should not exceed 1/10 of the layer thickness, and should be at least 20 times higher than the average roughness of the indented surfaces [46,47]. Another important limitation that needs to be considered for a reliable

nanoindentation test is represented by the tip geometry and radius used, which limits the good testing contact depths, hence, in our case, was necessary to obtain contact depths higher than 40 nm [48].

In view of these testing conditions, the hardness and reduced modulus of the investigated sample coatings were determined according to the Oliver–Pharr method using similar force–displacement curves to the ones presented in Figure 3a. As it can be seen, these curves were obtained in both elastoplastic (CoCr substrate) and nearly elastic regimes (TiCN and TiSiCN coatings) [49], which were used to obtain the mechanical parameters of the samples. Note that all experimental force–displacement curves were qualitatively similar for the TiCN and TiSiCN coatings, providing indentation depth values smaller than 130 nm and exhibiting well-defined edges of the residual indent impressions (inset in Figure 3a). Moreover, the relatively low surface roughness in the investigated regions of the samples (~5 nm) was confirmed by performing several scans on 4 $\mu m^2$ areas using the same Berkovich tip. It has to be noted that this roughness corresponds to the flat areas between the microparticles found on the surface and it is therefore much smaller than the values reported for 4000 $\mu m$ length surface scans. The resulting H = 7 ± 0.2 GPa and E = 97 ± 1.1 GPa values for the CoCr substrate are much smaller than the ones obtained for the investigated coatings. The hardness of the coatings significantly increased from 36.6 ± 2.9 GPa for the TiCN coating to 47.4 ± 1 GPa for TiSiCN, despite the small amounts of Si added into the TiCN layer structure (Figure 3b), while the effective modulus changed from 277 ± 8 GPa to 310 ± 2 GPa. The hardness enhancement originates from the grain size effect (H~$d^{-1/2}$, according to the Hall–Petch strengthening mechanism), since the TiSiCN coatings have smaller grain sizes. The hardness and modulus values were consistent with the previous results [50,51]. This implies that the condition H/E > 0.1 [52] was fulfilled for both coatings, as the calculated values for TiCN and TiSiCN were about 0.13 and 0.15, respectively. As the TiCN structure was modified, the sharp increase in the $H^3/E^2$ ratio (resistance to plastic deformation) from 0.63 to 1.1, testifies to the superior toughness of the TiSiCN layer [53], and it may be ascribed to the beneficial effect of the Si content. The $H^3/E^2$ parameter was considered to be a parameter sensitive to the tribological and corrosion properties of the materials [54].

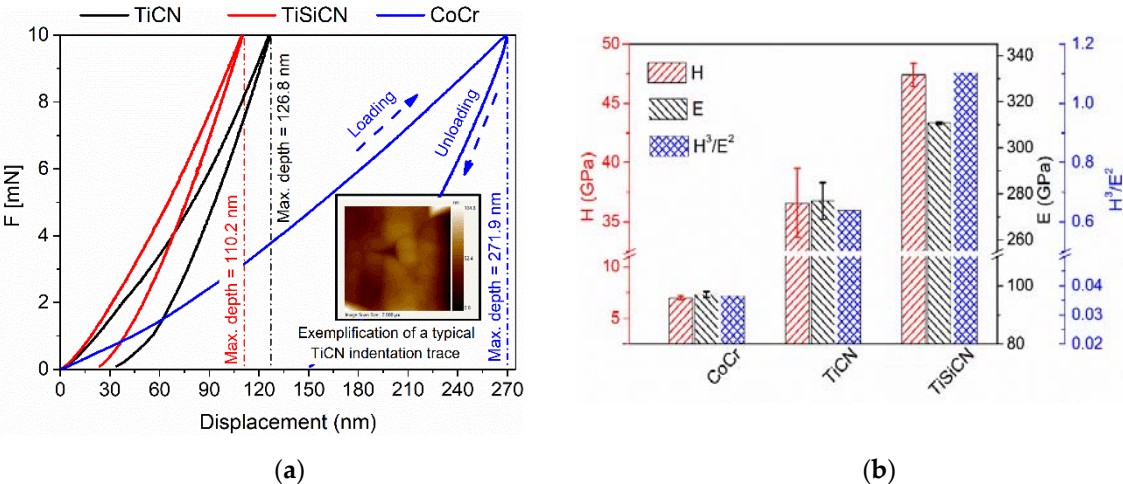

**Figure 3.** (**a**) Experimental force–displacement curves of the CoCr substrate, TiCN and TiSiCN coatings and (**b**) averaged hardness and reduced Young's modulus values.

### 3.5. In Vitro Corrosion Investigations

#### 3.5.1. Potentiodynamic Polarization

The open circuit potential ($E_{OC}$) evolution after 1 h of immersion and the potentiodynamic curves of the investigated systems are presented in Figure 4. During the immersion, the $E_{OC}$ of the TiCN and TiSiCN thin films showed a steady evolution, while the substrate slightly changed its value for half of the time, reaching a stable evolution at the end of the test. The TiSiCN coatings exhibited a positive

$E_{OC}$ value (54 mV) compared with the TiCN coatings, indicating that the addition of Si had a positive effect on corrosion behavior.

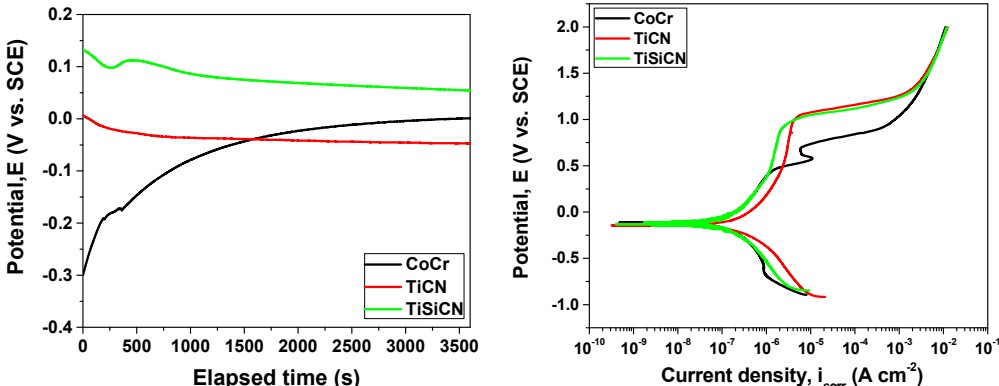

**Figure 4.** (**a**) Open circuit potential evolution in time and (**b**) potentiodynamic curves of the investigated systems.

The main electrochemical parameters of the investigated specimens calculated based on the Tafel and potentiodynamic curves are presented in Table 4 (all the presented potentials are relative to the SCE value). Polarization resistance ($R_p$) was determined from linear polarization measurements as the slope of the linear region of the $\Delta E$–$\Delta i$ curve near $E_{corr}$. Corrosion potential ($E_{corr}$), anodic ($\beta_a$) and cathodic ($\beta_c$) slopes were estimated from Tafel plots. The corrosion current density was also calculated based on one form of the Stern–Geary equation (Equation (1)), based on previously determined parameters.

$$\frac{1}{R_p} = \left(\frac{\Delta i}{\Delta E}\right)_{E_{corr}} = 2.3\left(\frac{\beta_a + |\beta_c|}{\beta_a\,|\beta_c|}\right) i_{corr} \tag{1}$$

considering the $E_{corr}$ parameter, the most electropositive value was demonstrated by the TiSiCN coating ($E_{corr\,TiSiCN} = -14$ mV), as well as the smallest value of corrosion current density ($i_{corr\,TiSiCN} = 49.6$ nA cm$^{-2}$). Taking into account the polarization resistance, the highest value was observed in the case of the CoCr substrate, closely followed by the TiSiCN coating ($R_{p\,TiSiCN} = 425$ kΩ·cm$^2$).

**Table 4.** The main corrosion parameters of the investigated specimens.

| Substrate/ Coating | $E_{oc}$ (mV) | $R_p$ (kΩ cm$^2$) | $E_{corr}$ (mV) | $\beta_a$ (mV dec$^{-1}$) | $\beta_c$ (mV dec$^{-1}$) | $i_{corr}$ (nA cm$^{-2}$) |
|---|---|---|---|---|---|---|
| CoCr | 1 | 455 | −36 | 516 | 447 | 198.2 |
| TiCN | −48 | 216 | −103 | 517 | 288 | 233.0 |
| TiSiCN | 54 | 425 | −14 | 271 | 269 | 49.6 |

3.5.2. Electrochemical Impedance Spectroscopy (EIS)

It was observed that the electrochemical performance of the investigated specimens was different as a function of their composition. For comparison, Nyquist and Bode plots for the investigated specimens are presented in Figure 5.

The electrochemical parameters were obtained by fitting the data with an equivalent circuit, which took into consideration the phenomenon at the interface of each investigated system with the testing electrolyte (inset Figure 5). $R_{el}$ represents the electrolyte resistance, CPE$_{layer}$ represents the coating capacitance, R$_{pore}$ represents the resistance associated with the current flow through the pores generated by the coatings' defects and CPE$_{dl}$ is a double layer capacitance in parallel with a charge transfer resistance - R$_{ct}$. CPE was used instead of a capacitor due to the non-ideal character of the

working electrode. The physical interpretation of a circuit that has a constant phase element (for a better quality fit) depends on the value of $\alpha$. If the $\alpha$ parameter is 1, then the CPE can be modeled as a capacitor. Since after the fitting, the $\alpha$ parameter showed values less than 1 in both cases (i.e., the $\alpha$ layer and $\alpha_{dl}$), a CPE was used. This can be due to possible deviations from the ideal dielectric behavior and it is usually related to the surface inhomogeneity [55]. According to Hirschorn et al. [56], these deviations arise either from different properties along the surface of an electrode (e.g., roughness), or properties normal for the surface (e.g., thickness).

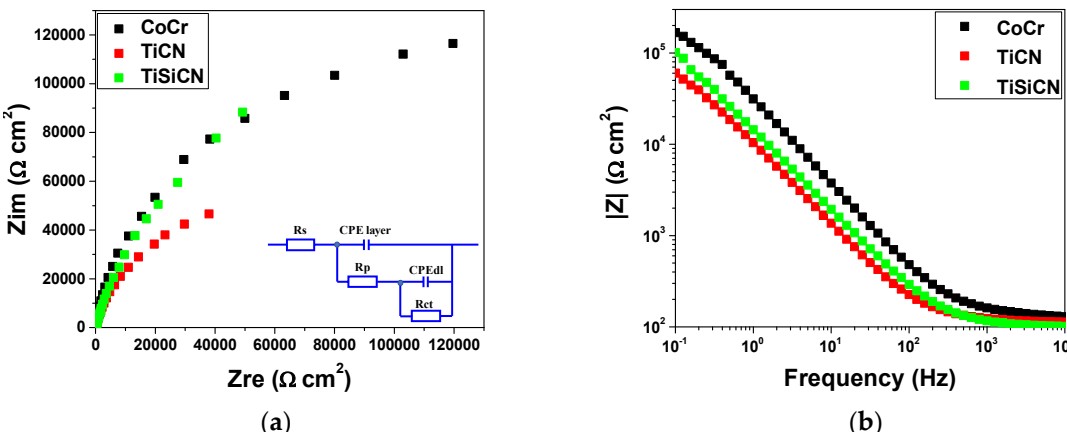

**Figure 5.** (**a**) Nyquist plot (electrical circuit used for the fitting procedure included) and (**b**) Bode plot of the investigated specimens.

The electrochemical parameters of the investigated systems are presented in Table 5. It can be noticed the low value of the $\chi^2$ parameter, which is an indication of an excellent agreement between the experimental data and those simulated by the equivalent circuit.

**Table 5.** The fitting results of EIS curves for the investigated systems.

| Substrate/ Coating | $R_{el}$ ($\Omega \cdot cm^2$) | $Q_{layer}$ ($\mu Fs^{(\alpha-1)}$ $cm^{-2}$) | $\alpha$ Layer | $R_{pore}$ ($k\Omega \cdot cm^2$) | $Q_{dl}$ ($\mu Fs^{(\alpha-1)}$ $cm^{-2}$) | $\alpha$ dl | $R_{ct}$ ($k\Omega \cdot cm^2$) | $\chi^2$ |
|---|---|---|---|---|---|---|---|---|
| CoCr | 127 | 1.94 | 0.96 | 88 | 3.66 | 0.91 | 243,830 | $13 \times 10^{-3}$ |
| TiCN | 113 | 16.19 | 0.87 | 216 | 2.19 | 0.99 | 112,000 | $3 \times 10^{-3}$ |
| TiSiCN | 101 | 12.38 | 0.87 | 316 | 1.21 | 0.96 | 262,750 | $5 \times 10^{-3}$ |

Taking into account the fitting results for the investigated systems, it can be noticed that the highest pore-associated resistance was obtained for TiSiCN, while the CoCr substrate showed low values. It was stated that CoCr alloys form a passive layer at the surface, which is mainly based on Cr(III), and smaller amounts of $Cr(OH)_3$, Co and Mo oxides [57]. The fitted values associated to the CoCr specimen showed that the formed layer is not as compact and protective as the TiCN and TiSiCN coatings.

The $Q_{dl}$ parameter, which is representative of the substrate–electrolyte interface, indicated a better protection of the deposited/formed layer in the following order: TiSiCN > TiCN > CoCr. Thus, the best protection after immersion in 90 % DMEM + 10 % FBS was observed for the TiSiCN coatings, with the best capacitive character, indicated by the low value of $Q_{layer}$ and the highest value of $R_{ct}$.

Considering $\alpha_{layer}$, it can be observed that for CoCr, the CPE used for fitting the obtained data were the closest to a capacitor, since in this case, $\alpha_{layer} = 0.96$. This could be due to the low roughness measured before the corrosion as compared to the other investigated specimens (Table 3). Similar $\alpha$ layer values were obtained for the coatings and the time-constant dispersion is ascribed to the similar values of roughness. Going deeper, at the interface between the coating and the substrate, another double layer is formed. $Q_{dl}$ and $\alpha_{dl}$ can give an indication of the compactness of the deposited/formed layer and the electrolyte ingress through the defects, which can create pathways for the electrolyte to

reach the substrate [58]. It can be observed that even though $\alpha_{dl}$ was higher for TiCN, showing an almost defect-free structure, TiSiCN was the one showing better values of $Q_{dl}$ and $R_{ct}$.

### 3.6. Morphology and Roughness after Corrosion

SEM images after the corrosion tests are presented in Figure 6. It is worth noting that the uncoated substrate was more affected by the corrosion than the coated surfaces. The destruction of the protection layer can be seen on the coated samples, after performing the corrosion tests. Regarding the TiCN coatings, there were various corrosion products on the surface, indicating that this surface was affected by the corrosive solution. Moreover, the coating was partially destroyed in some areas, with the CoCr substrate being visible. The TiSiCN surface also had corrosion products, but there were less compared with the TiCN surface, indicating better anticorrosive properties. All surfaces were affected by the corrosive solution, but TiSiCN was less damaged.

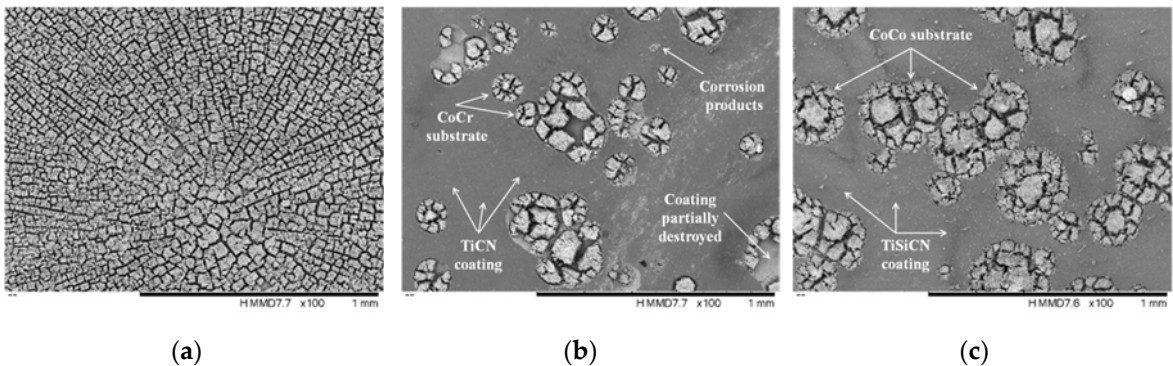

|       (a)       |       (b)       |       (c)       |

**Figure 6.** SEM of the investigated (**a**) CoCr substrate and (**b**) TiCN and (**c**) TiSiCN coatings.

The main roughness parameters of the investigated surfaces are presented in Table 6. Comparing these results with the values obtained before the corrosion tests (Table 3), it can be said that all surfaces were significantly damaged after the corrosion tests. The $R_a$ of the uncoated substrates increased from 46.9 ± 5.9 nm to 1342.9 ± 192.4 nm. A significant increase in $R_a$ roughness was found for the TiCN coatings after corrosion (15584.3 ± 7462.8 nm), indicating a major deterioration of the coatings after the corrosion tests. The TiSiCN coatings were also affected by the corrosive process, but they have finer irregularities than TiCN, demonstrating that the addition of Si led to an enhancement of anticorrosive properties. All surfaces showed a negative value of $S_k$ after the corrosion tests, signifying that the surfaces were characterized by many valleys formed during the corrosive processes. The TiSiCN surface had fewer valleys that the TiCN surface, as shown by the smaller absolute value of $S_k$.

**Table 6.** Main roughness parameters of the investigated specimens after corrosion tests.

| Sample | CoCr | TiCN | TiSiCN |
|:---:|:---:|:---:|:---:|
| $R_a$ (nm) | 1342.9 ± 192.4 | 15,584.3 ± 7462.8 | 6297.9 ± 1598.3 |
| $R_q$ (nm) | 1758.2 ± 237.2 | 19,896.6 ± 6410.9 | 7480.5 ± 1567.1 |
| $S_k$ | −0.1 ± 0.2 | −1.2 ± 0.8 | −0.7 ± 0.4 |
| Kurtosis | 2.4 ± 0.3 | 1.8 ± 0.4 | 0.7 ± 0.3 |

## 4. Discussions

The nature of the electrolyte, scanning rate, temperature, impurities, anode material and surface state of the samples are a part of the parameters, which can influence the electrochemical reactions. For example, the surface texture of the working electrode (investigated sample) is one of the most important parameters, which has an influence on the Tafel slopes, and consequently on the corrosion rate. Surface roughness has an important effect on general or pitting corrosion and the nucleation of

metastable pitting. Skewness and kurtosis were used to identify the corrosion mechanism. Reid et al. have patented a method and apparatus for the identification of corrosion in metal objects and defining typical values of skewness and kurtosis for the identification of corrosion mechanism [59]. The pitting mechanism appears when the $S_k < -2$. In all our cases, the $S_k$ had values greater than $-2$, indicating that a general corrosion mechanism can be found for all investigated surfaces. Regarding the kurtosis, if the value is $< 3$, general corrosion can be observed, while for kurtosis $> 3$, pitting corrosion can be found. Considering this, it can be seen that for all investigated surfaces, a general corrosion mechanism has been identified. If the surface is rough, then a larger area could contribute to the increase in the corrosion current or to the corrosion rate. Therefore, a decreased surface roughness will lead to a better corrosion resistance [60]. According to this statement, the TiCN-coated surface was rougher than TiSiCN, and this is probably the reason for its better corrosion resistance. For the uncoated CoCr, the roughness is not a factor which has a major influence on the corrosion resistance. Compared to the coated samples, the CoCr uncoated substrate had a smaller roughness, while its corrosion resistance was worse than for both coatings. Clearly, the surface roughness affects the corrosive behavior of materials (i.e., metals, alloys, coatings) and the nature of its effects (increase or decrease in the degree of corrosion) depends on the nature of the material.

To the best of our knowledge, there is no direct relation between hardness and corrosion resistance. Hard coatings can be subjected to surface microcracks, and then a localized penetration of corrosive solution will take place, leading to a galvanic cell, which accelerates the corrosive process. Hardness is important for load-bearing implants because a hardened material can have the ability to withstand wear. Taking into account the results of the present study, TiSiCN was harder than CoCr and the TiCN coatings and was more adequate for the proposed application.

For load-bearing implants, resistance to plastic deformation is an important factor and it can be described by the $H^3/E^2$ ratio. Moreover, a material with a high $H^3/E^2$ ratio resists plastic deformation during low load contact events and exhibits a higher yield strength [61,62]. It is also generally accepted that the H/E ratio can be considered an important indicator of a good wear resistance of the surface [63–65]. Thus, the improvement of the H/E ratio and, consequently, of the resistance to plastic deformation ($H^3/E^2$ ratio) of the load-bearing implant may offer advantages, such as less surface damage and increased durability. In this study, the TiSiCN coatings have an $H^3/E^2$ ratio equal to 1.1, which is higher than the one for TiCN (0.63), indicating that TiSiCN has a superior toughness and it can offer a better resistance to plastic deformation and good wear resistance.

The addition of Si to TiCN coatings leads to a grain refinement, and the crystallite size (d) was decreased to 16.4 nm in the case TiCN and to 14.6 nm in the case of TiSiCN. The formation of new defects, especially dislocations, is also responsible for the reduction in the crystallite size. The strain in the TiSiCN coatings ($\varepsilon = 0.012$) was lower compared to the TiCN coatings ($\varepsilon = 0.053$). The reason for this decrease could be attributed to different factors. One reason could be due to the addition of Si, which has atomic radii (0.111) smaller than that of Ti (0.146 nm), leading to a disorder of the crystal lattice, which is also evident by XRD diffraction (peaks were shifted when compared with the TiCN standard). The second reason could be attributed to amorphous phases in which Si can be found (Si, $SiN_x$, SiCN) or C=C phases at grain boundaries, which are detrimental for crystallite development. TiSiCN has a higher C content than TiSiC. It is difficult to separate these factors and to know their contribution. However, the crystalline disorder becomes more pronounced by an increase in carbon content, which is also suggested by the decrease in the crystallite size and by the decrease in microstrain. Moreover, Franceschini et al. reported a strong dependence of stress on the nitrogen content in a-C:H films; at a low N content, the stress is high [66]. This effect is difficult to see in our coatings, because the N content is reduced after the addition of Si, but it is a minor reduction. This result can also have a major influence on the corrosion resistance of TiSiCN. This coating probably presents fewer defects and it is more compact than TiCN.

When the crystallite size decreases, the corrosion current density decreases and polarization resistance increases, which means that the corrosion resistance of the coatings increases with decreasing

grain size. Thus, the TiSiCN coatings, which have the smallest crystalline size, were more resistant to corrosive attack. The dependence of corrosion resistance on crystallite size can be ascribed to the BOLS mechanism [67]. In the grain boundaries, there are undercoordinated atoms with lowered residual cohesive energy which possess high energy, these atoms exist in unstable states and an increase in their percent will lead to an increase of corrosion resistance [68]. This finding is also sustained by the strain $\varepsilon$ value. In both cases, the strain was low, but after the addition of Si, the strain was significantly reduced. When the strain decreases, the corrosion resistance of the coatings increases. Thus, the correlation between high corrosion resistance and low strain and small crystallites can be explained in terms of the "bond-order-length-strength correlation mechanism", meaning that the undercoordinated atoms found on the surface or in grain boundaries take the responsibility of the good corrosion resistance. In the current paper, along with the addition of Si, the $Q_{dl}$ parameter was also decreasing, and this result could be due to a smaller crystallite size obtained by the TiSiCN coating. Thus, the decreased generation of defects, in the case of this coating, had a beneficial effect on the protective properties. It was shown that defects within a structure can cause localized corrosion at the coating–substrate interface, due to the electrolyte ingress [69]. In addition, $R_{pore}$ indicated that the resistivity of the electrolyte in the pores had the highest value in this case, which can be also correlated with the lack of defects. The $\alpha$ values ranged from about 0.87 to 0.90, and the deviation from an ideal capacitor was ascribed to differences in roughness, as was shown. The dependence between roughness, capacitance and associated $\alpha$ values was demonstrated [69], although there are also some other factors which can be influences, such as thickness and the dielectric constant of the material.

## 5. Conclusions

The present study aimed to find a new and improved possible solution for load-bearing implants. For this purpose, titanium-based carbonitrides with and without the addition of Si were investigated and compared. Both coatings obtained by the cathodic arc deposition method had an almost stoichiometric structure, being solid solution, which consisted of a mixture of TiC and TiN, with a face-centered cubic (FCC) structure. The crystallite size decreased with the addition of Si into the TiCN matrix, the crystallite size of TiCN was 16.4 nm, while for TiSiCN it was 14.6 nm. Both coated surfaces exhibited a uniform coverage, with some microparticles from the ion sputtering and ejection of the particles during the deposition process. After the addition of Si into TiCN, the $R_a$ roughness values decreased, indicating a beneficial effect of Si.

All investigated surfaces had positive skewness, which is adequate for load-bearing implants, which work in corrosive environments. The hardness of the TiCN coating was 36.6 ± 2.9 GPa and it was significantly increased to 47.4 ± 1 GPa when small amounts of Si were added into the TiCN layer structure, while the elastic modulus was increased from 277 ± 8 GPa to 310 ± 2 GPa. A significant increase in resistance to plastic deformation ($H^3/E^2$ ratio) from 0.63 to 1.1 was found after the addition of Si into the TiCN matrix.

The most electropositive value of corrosion potential was found for the TiSiCN coating (−14 mV), as well as the smallest value of corrosion current density (49.6 nA cm$^2$), indicating good corrosion resistance. The TiSiCN coating exhibited the best protection after immersion in 90% DMEM + 10% FBS, the best capacitive character, indicated by the low value of $Q_{dl}$, and the highest resistance through the pores generated by the defects of the coatings and the electrolyte ingress, indicated by $R_{pore}$ and $R_{ct}$.

According to the conducted research, TiSiCN coatings have shown good mechanical properties and high corrosion resistance and are a good alternative for the coating of load-bearing implants.

**Author Contributions:** Conceptualization, A.V. and C.V.; methodology, A.V.; investigation, M.D., P.S., I.G.S., and I.P.; resources, A.V.; data curation, I.P.; writing—original draft preparation, M.D., I.P., and P.S.; writing—review and editing, C.V. and A.V.; supervision, A.V. All authors have read and agreed to the published version of the manuscript.

**Funding:** The present work was supported under a grant of the Romanian National Authority for Scientific Research, CNCS—UEFISCDI, project No. PN-III-P1-1.2-PCCDI-2017-0239/60PCCDI 2018, within PNCDI III. The

EDS, SEM, XRD and nanoindentation results were acquired using the systems purchased by the infrastructure project INOVA-OPTIMA SMIS code 49164, contract No. 658/2014. A part of work is also supported by PROINSTITUTIO Project–contract No. 19PFE/17.10.2018.

**Conflicts of Interest:** The authors declare no conflict of interest. The funders had no role in the design of the study; in the collection, analyses, or interpretation of data; in the writing of the manuscript, or in the decision to publish the results.

## Nomenclature

**Roman Symbols**

| | |
|---|---|
| $CPE_{dl}$ | Constant phase element that models the behavior of a double layer |
| $CPE_{layer}$ | Constant phase element that models the behavior of a layer |
| d | Crystallite size |
| E | Elastic modulus |
| $E_{corr}$ | Corrosion potential |
| $E_{OC}$ | Open circuit potential |
| F | Force |
| H | Hardness |
| $i_{corr}$ | Corrosion current density |
| $Q_{dl}$ | Capacitance of the double layer |
| $Q_{layer}$ | Capacitance of the layer |
| $R_a$ | Arithmetic average of the roughness profile |
| $R_{ct}$ | Charge transfer resistance |
| $R_{el}$ | Electrolyte resistance |
| RMS | Root mean square |
| $R_p$ | Polarization resistance |
| $R_{pore}$ | Resistance associated to the current flow through the pores |
| $R_q$ | Root means square average of the roughness profile |
| $S_k$ | Skewness |
| T(hkl) | Texture coefficient |
| Z | Impedance |
| $Z_{im}$ | Imaginary part of impedance |
| $Z_{re}$ | Real part of impedance |

**Greek Symbols**

| | |
|---|---|
| $2\theta$ | Angle between incident beam and reflected beam |
| $\alpha$ dl | Exponent equaling 1 for a capacitor characteristic to the double layer |
| $\alpha$ layer | Exponent equaling 1 for a capacitor characteristic to the layer |
| $\beta_a$ | Anodic Beta coefficient of Tafel slope |
| $\beta_c$ | Cathodic Beta coefficient of Tafel slope |
| $\varepsilon$ | Strain |
| $\chi^2$ | Chi-square statistic distribution |

**Acronyms**

| | |
|---|---|
| CE | Counter electrode |
| CVD | Chemical vapor deposition |
| DMEM | Dulbecco's Modified Eagle's Medium |
| EDS | Energy dispersive spectrometry |
| EIS | Electrochemical impedance spectroscopy |
| FBS | Fetal bovine serum |
| FCC | Face centered cubic structure |
| JCPDS | Joint committee on powder diffraction standards |
| PVD | Physical vapor deposition |
| RE | Reference electrode |
| SCE | Saturated Calomel electrode |
| SEM | Scanning electron microscopy |
| XRD | X-Ray Diffraction |

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
