# Peer review of "Improvement of CoCr Alloy Characteristics by Ti-Based Carbonitride Coatings Used in Orthopedic Applications"

_coatings, doi:10.3390/coatings10050495_

Round 1

Reviewer 1 Report

Dear Authors,

First of all, I would like to express my positive opinion about the work. The presented results are valuable and for sure worth for publication. In my opinion, the presented manuscript meets all requirements for publication. The paper demonstrates excellent TiSiCN properties in further applications. As a minor comment I would like to recommend revision of the manuscript and:

a) add the nomenclature (generally all is explained in the text, but sometimes it is not easy to follow it...)

b) Please use always passive voice whole the manuscript - for instance in conclusions section is //we...

Author Response

Ref: coatings-789756_R1
Title: Improvement of CoCr alloy characteristics by Ti based carbonitrides coatings used in orthopedic applications

Dear Editor,

Thank you for your note and the reviewer’s comments on our manuscript. We would like to show our great gratitude to the reviewer for the useful comments and constructive suggestions on our manuscript, which do help us significantly improve the quality of the current paper. All the review comments are appreciated. We do benefit a lot from the suggestions/ comments to improve the quality of our manuscript. We have revised our manuscript accordingly. The revision of the paper was highlighted by the blue coloured font. Detailed and point-to-point response to the reviewer’s comments is summarized below.

Here, we re-submit a new version of our manuscript which has been checked and modified after our careful referring to the reviewers’ comments. Meanwhile, efforts were also made to improve the English of the paper. We hope all of these changes will make this manuscript accepted by reviewers. Thank you for your kind consideration.

Best regards,

Alina Vladescu

Reviewer 1

Top of Form

Dear Authors,

First of all, I would like to express my positive opinion about the work. The presented results are valuable and for sure worth for publication. In my opinion, the presented manuscript meets all requirements for publication. The paper demonstrates excellent TiSiCN properties in further applications. As a minor comment I would like to recommend revision of the manuscript and:

a) add the nomenclature (generally all is explained in the text, but sometimes it is not easy to follow it...)

Thank you for this observation. A list of abbreviations was added at the end of the manuscript (pages 13-14).

b) Please use always passive voice whole the manuscript - for instance in conclusions section is //we...

Following this suggestion, the whole paper was revised and the English grammar was modified.

 Bottom of Form

Reviewer 2 Report

This paper studied the CoCr alloy characteristics by Ti based carbonitrides coatings used in orthopedic applications. TiCN and TiSiCN coatings were prepared by the cathodic arc deposition method and were analyzed as possible solution for load bearing implants. This paper is written well and can be accepted after the following revisions:

1). Keywords:

Please do not use the abbreviations in the keywords.

2). Introduction:

Please compare the CoCr alloy with other alloys. What are the other applications of CoCr alloy?

3). Materials and Methods:

Please add the detailed information of the materials used in the experiment.

Line 104, please change "conditions" to "Conditions". English writing should be improved in the revised paper.

4). Results:

Please show the AFM results of the surface roughness.

5). Conclusions:

This section needs to be rewritten. Please firstly list provide the main objectives in this work. The key findings should be merged. 

6). References:

More recent papers published in the last 3 years should be added.

Author Response

Ref: coatings-789756_R1
Title: Improvement of CoCr alloy characteristics by Ti based carbonitrides coatings used in orthopedic applications

Dear Editor,

Thank you for your note and the reviewer’s comments on our manuscript. We would like to show our great gratitude to the reviewer for the useful comments and constructive suggestions on our manuscript, which do help us significantly improve the quality of the current paper. All the review comments are appreciated. We do benefit a lot from the suggestions/ comments to improve the quality of our manuscript. We have revised our manuscript accordingly. The revision of the paper was highlighted by the blue coloured font. Detailed and point-to-point response to the reviewer’s comments is summarized below.

Here, we re-submit a new version of our manuscript which has been checked and modified after our careful referring to the reviewers’ comments. Meanwhile, efforts were also made to improve the English of the paper. We hope all of these changes will make this manuscript accepted by reviewers. Thank you for your kind consideration.

Best regards,

Alina Vladescu

Reviewer 2

This paper studied the CoCr alloy characteristics by Ti based carbonitrides coatings used in orthopedic applications. TiCN and TiSiCN coatings were prepared by the cathodic arc deposition method and were analyzed as possible solution for load bearing implants. This paper is written well and can be accepted after the following revisions:

1). Keywords: Please do not use the abbreviations in the keywords.

We would like to thank the evaluator for reviewing the paper and for the comments. The authors revised the manuscript and all changes are in blue color.

The Keywords section was changed by removing all abbreviations.

2). Introduction: Please compare the CoCr alloy with other alloys. What are the other applications of CoCr alloy?

More details about the applications of CoCr alloys were added in the manuscript and comparisons with other alloys were made. Thus, an additional paragraph (lines 40-50) was inserted in the Introduction section as follows:

CoCr alloys have been used also as screws in trauma plating systems. In this case, a low osseointegration, compared to Ti alloys, could facilitate an easier removal after the healing of the bone fracture. Due to its higher strength CoCr alloy was used also for idiopathic scoliosis applications, where the results proved to be better in the case of CoCr alloy compared to stainless steel (SS) and Ti alloy [4]. Another application was the use of CoCr alloy in implants used for correction of spine deformities due to high rigidity of the used CoCr rod compared to SS and Ti based ones [5,6]. CoCr was found to be well adapted also in dentistry for its good castability, especially the wrought alloys. Guide wires, clips, orthodontic arch wires, catheters are among the main applications in this field [7,8]. Thus the decreased corrosion resistance of SS and the low wear resistance of Ti alloys make it difficult to replace CoCr alloys in a wide range of applications.”

3). Materials and Methods: Please add the detailed information of the materials used in the experiment.

Details about cathodes supplier, reactive gases used and associate purity were added in the manuscript. Specifications about CoCr alloy type were also included.

Line 104, please change "conditions" to "Conditions". English writing should be improved in the revised paper.

In the manuscript ‘’conditions” was replaced by "Conditions" and the English writing was improved in the resubmitted version.

4). Results: Please show the AFM results of the surface roughness.

In this study, the roughness was measured using a surface profilometer. The presented data are obtained by five line-scans, each on a length of 4000 μm. The characteristic parameters (Ra, Rq and Sk) were determined using a scanning line profile which was recorded with respect to a fixed scan length and duration for obtaining a good horizontal resolution. The authors did not obtain conclusive results after performing preliminary AFM measurements, since the sample was not fitted for this type of analysis: the roughness exceeded the maximum of Z range specific to the scanner which is ranging between ± 3 μm. Also, it was more relevant to assess the surface roughness over large distances as long as the envisaged application is addressed to the load bearing implants.

5). Conclusions: This section needs to be rewritten. Please firstly list provide the main objectives in this work. The key findings should be merged. 

Thank you for your helpful advice. The Conclusion section was entirely rewritten and objectives were added.

6). References: More recent papers published in the last 3 years should be added.

The manuscript was modified according to the suggestion. Therefore, a total of 12 new references were added as compared to the first submitted version of the manuscript, from which a number of 4 papers were published in the last 3 years.

Reviewer 3 Report

The publication entitled, ‘ Improvement of CoCr alloy characteristics by Ti based carbonitrides coatings used in orthopedic applications’, by Dinu et al gives an experimental account of the corrosion resistance, hardness and elasticity of Ti(Si)CN based coatings on CoCr used as implants. The work is interesting but there are certain discrepancies in the data and their interpretation. I would really recommend that the authors consider the following suggestions:

1). The authors mention on line 49, ‘The main problem of these debris is their high size,’, what about Cr or Co ion release into blood or tissue? What effects do they have?

2). Please explain on line 156, ‘strain is significantly reduced.’ as a function of crystallite size

3). Please explain further why this happens in line 160, ‘Both coatings exhibited the unit cell parameters less than that of used standard’

4). In figure 3, why wasn’t hardness performed for CoCr?

5). In the XRD how do you know that Si was integrated into TCN? This is all the more important to understand as the lattice parameter reduces after the addition of Si. Why do the authors show increased lattice parameter on Si addition when the (111) peak of TSCN is shifted to higher 2ÆŸ. If they are immiscible i.e TiN and TiC, do you observe secondary phases? There is no unit for lattice parameter in lines 158, 159.

6). In the SEM images of figure 2. The particulate formation composition on the surface should be checked with EDX for both coatings. Since the surfaces have been polished, I would like the authors to check if they are secondary phases.

7). This aggregation of secondary phases will have an effect on the nanoindentation and should be discussed.

8). SEM images of figure 6 should be done at a lower magnification so that we have a better and larger overview of the surface. For the moment considering figure 6(c) with a ‘cut-out’ corrosion area, it is very difficult to compare it to 6(b). All 3 images should be captured at the same lower magnification.

9). Please do not provide the conclusion as bullets. Please stick to the standard format of a descriptive text.

I would like the authors to clarify the above points before I can accept this paper.

Author Response

Ref: coatings-789756_R1
Title: Improvement of CoCr alloy characteristics by Ti based carbonitrides coatings used in orthopedic applications

Dear Editor,

Thank you for your note and the reviewer’s comments on our manuscript. We would like to show our great gratitude to the reviewer for the useful comments and constructive suggestions on our manuscript, which do help us significantly improve the quality of the current paper. All the review comments are appreciated. We do benefit a lot from the suggestions/ comments to improve the quality of our manuscript. We have revised our manuscript accordingly. The revision of the paper was highlighted by the blue coloured font. Detailed and point-to-point response to the reviewer’s comments is summarized below.

Here, we re-submit a new version of our manuscript which has been checked and modified after our careful referring to the reviewers’ comments. Meanwhile, efforts were also made to improve the English of the paper. We hope all of these changes will make this manuscript accepted by reviewers. Thank you for your kind consideration.

Best regards,

Alina Vladescu

Reviewer 3

Top of Form

The publication entitled, ‘Improvement of CoCr alloy characteristics by Ti based carbonitrides coatings used in orthopedic applications’, by Dinu et al gives an experimental account of the corrosion resistance, hardness and elasticity of Ti(Si)CN based coatings on CoCr used as implants. The work is interesting but there are certain discrepancies in the data and their interpretation. I would really recommend that the authors consider the following suggestions:

We would like to thank the evaluator for reviewing the paper and for the comments. The authors revised the manuscript and all changes are in blue color.

1). The authors mention on line 49, ‘The main problem of these debris is their high size,’, what about Cr or Co ion release into blood or tissue? What effects do they have?

Thank you for your helpful advice. More details of the effect of Cr or Co ion release into blood or tissue were added in the manuscript (lines 63-66).

2). Please explain on line 156, ‘strain is significantly reduced.’ as a function of crystallite size

a). Williamson–Hall (W–H) method considers that the origin of the lattice strain is attributed mainly to the lattice expansion or lattice contraction in the nanocrystals due to size confinement, because the atomic arrangement gets slightly modified due to size confinement. On the other hand, many defects also get created at the lattice structure due to the size confinement and this in turn results in the lattice strain. The value of the lattice parameter is influenced by both chemical composition and strain (stress) in the coatings. For example, a compressive residual stress induces a lattice expansion, while the presence of atoms with different radii and/or changes in stoichiometry determines specific modifications of the lattice constant [G. Abadias, Surface and Coatings Technology, 202 (2008), pp. 2223-2235].

b). The strain-induced broadening due to crystal imperfection and distortion.

It is reasonable to believe that the observed difference in strain of TiSiCN coatings is related to the amorphous phases in which Si can be found (Si, SiNx, SiCN) or C=C phase at grain boundaries, being detrimental for crystallite development. These effects will be goal of other paper, in which other complex analysis such as TEM and XPS will be carried out. This text was added in the manuscript.

c). An additional paragraph (lines 398-412) was inserted in the Discussion section as follows:

The presence of Si addition in TiCN coating leads to a grain refinement, and the crystallite size (d) decreased from 16.4 nm in the case TiCN to 14.6 nm in the case of TiSiCN. The formation of new defects, especially dislocations, is also responsible for the reduction of the crystallite size. The strain in TiSiCN coatings (e = 0.012) was lower compared to the TiCN coatings (e =0.053). The reason of this decrease could be attributed to different factors. One reason can be due to the Si addition, which has atomic radii (0.111) less that of Ti (0.146 nm), leading to a disorder to the crystal lattice, being also evident by XRD diffraction (peaks were shifted compared with TiCN standard). Second reason can be attributed to amorphous phases in which Si can be found (Si, SiNx, SiCN) or C=C phase at grain boundaries, being detrimental for crystallite development. TiSiCN has a bit high C content than TiSiC. It is difficult to separate these factors and to know their contribution. However, the crystalline disorder is more pronounced by increasing of carbon content, being also suggested by the decrease of the crystallite size and by decreasing of microstrain. Moreover, Franceschini et al. reported a strong dependence of the stress on the nitrogen content in a-C:H films: at low N content, the stress is high [65]. This effect is difficult to be seeing in our coatings, because the N content is reduced after the Si addition, but it is a small reduction.”

3). Please explain further why this happens in line 160, ‘Both coatings e xhibited the unit cell parameters less than that of used standard’

Thank you for the observation. The authors corrected the sentence and it was replaced by “Both coatings exceeded the size of the standard unit cell parameters” (line 185).

4). In figure 3, why wasn’t hardness performed for CoCr?

Thank you for pointing this out. We modified figure 3 according to the reviewer suggestion by adding a typical force-displacement curve for CoCr substrate and its average values of hardness and modulus. Also, minor changes have been made in the 3.4 subsection.

5). In the XRD how do you know that Si was integrated into TCN? This is all the more important to understand as the lattice parameter reduces after the addition of Si. Why do the authors show increased lattice parameter on Si addition when the (111) peak of TSCN is shifted to higher 2ÆŸ. If they are immiscible i.e TiN and TiC, do you observe secondary phases? There is no unit for lattice parameter in lines 158, 159.

a). According to XRD, the peak (111) for both TiCN and TiSiCN coatings were shifted at towards lower Bragg angle compared with TiCN standard. To be observed this, we have added in the Fig.1, the peaks of TiCN standard (JCPDS 042-1488).

b). An additional paragraph (lines 164-167) was inserted in the manuscript as follows:

“When the diffraction peak is shifted towards lower Bragg angle, an increase of lattice constant is observed. This result is also in good agreement with those of Constantin et al. related to TiCN coatings [34]. It can be seen, that our TiCN is also shifted towards lower Bragg angle compared with TiCN standard, being related to the formation of more TiC phase. This finding is in accordance with the results reported by Wang et al. [35] regarding the TiSiCN coatings.”

c). Lattice parameters are presented in lines 183-185.  

6). In the SEM images of figure 2. The particulate formation composition on the surface should be checked with EDX for both coatings. Since the surfaces have been polished, I would like the authors to check if they are secondary phases.

a). The particulate formation is related to the energetic ejection of microparticles from the cathode surface, due to local melting at the interaction with the sputtering arc. The composition of the droplets is therefore close to the one of the cathode surface in the point and at the moment of droplet ejection. The droplets then adhere to the coating forming on the substrate and contribute to the increase of surface roughness, causing deterioration in the composition uniformity and the exfoliation of the coating. These droplets consist mainly of same composition as the coatings, with only minor differences. Please find below an image of EDS performed in different microdroplets found on TiSiCN surface. We have also added the elemental composition performed on large area (mapping results), showing that each individual droplet has only small deviation from the overall composition of the coating.

b). Please note that this is only a particular case, when only one cathode is used for the deposition, as in this study. When two or more cathodes are used for obtaining a multicomponent coating, it is possible that the droplets have indeed very different compositions, depending on their originating cathode (containing mainly the elements from that specific cathode).

7). This aggregation of secondary phases will have an effect on the nanoindentation and should be discussed.

There are no secondary phases in the coatings. However, it is hard to differentiate TiC and TiN since they have NaCl-type (B1) face-centered cubic (FCC) structure and the atom radius of C and N is similar which results in that N atoms are substituted by C atoms in the TiN structure or C atoms are substituted by N atoms in the TiC structure. Moreover, the indentation tests were performed only on the “smooth” coating surface, found in between the droplets. Therefore, the test conditions were chosen such as the droplets have no influence on the results.

8). SEM images of figure 6 should be done at a lower magnification so that we have a better and larger overview of the surface. For the moment considering figure 6(c) with a ‘cut-out’ corrosion area, it is very difficult to compare it to 6(b). All 3 images should be captured at the same lower magnification.

The SEM images were replaced in the manuscript and lower magnification micrographs for a larger overview of the surface were added.

9). Please do not provide the conclusion as bullets. Please stick to the standard format of a descriptive text.

Thank you for this remark. The Conclusion section was entirely rewritten according to the suggestion and objectives were added.

We hope all of these changes will make this manuscript accepted by reviewer. Thank you for your kind consideration.

Round 2

Reviewer 3 Report

The authors have replied satisfactorily to this reviewer and have carried out modifications.